# Accelerating Causal Inference and Feature Selection Methods through G-Test Computation Reuse

**DOI:** 10.3390/e23111501

**Published:** 2021-11-12

**Authors:** Camil Băncioiu, Remus Brad

**Affiliations:** Department of Computer Science and Electrical Engineering, Lucian Blaga University of Sibiu, 550024 Sibiu, Romania; remus.brad@ulbsibiu.ro

**Keywords:** Markov blanket, feature selection, causal inference, G-test, information theory, computation reuse

## Abstract

This article presents a novel and remarkably efficient method of computing the statistical G-test made possible by exploiting a connection with the fundamental elements of information theory: by writing the *G* statistic as a sum of joint entropy terms, its computation is decomposed into easily reusable partial results with no change in the resulting value. This method greatly improves the efficiency of applications that perform a series of G-tests on permutations of the same features, such as feature selection and causal inference applications because this decomposition allows for an intensive reuse of these partial results. The efficiency of this method is demonstrated by implementing it as part of an experiment involving IPC–MB, an efficient Markov blanket discovery algorithm, applicable both as a feature selection algorithm and as a causal inference method. The results show outstanding efficiency gains for IPC–MB when the G-test is computed with the proposed method, compared to the unoptimized G-test, but also when compared to IPC–MB++, a variant of IPC–MB which is enhanced with an AD–tree, both static and dynamic. Even if this proposed method of computing the G-test is presented here in the context of IPC–MB, it is in fact bound neither to IPC–MB in particular, nor to feature selection or causal inference applications in general, because this method targets the information-theoretic concept that underlies the G-test, namely conditional mutual information. This aspect grants it wide applicability in data sciences.

## 1. Introduction

Statistical tests of independence are mathematical tools used to determine whether two random variables, recorded in a data set as features, are more likely to be independent of each other, as opposed to being dependent, given a significance threshold. They are widely used in the analysis of experimentally collected data and relied upon to determine relationships between observed phenomena.

One major drawback of independence tests is their lack of causal information. In other words, such a test will not show which of the two tested features influences the other; it can only establish whether they are independent or not. However, it is often the causal information that is desired by researchers [1].

In order to extract causal information from a data set, the methods of causal inference may be applied. Causal inference is the field concerned with methods which build causal models from data sets. Such models encode causal relationships between the features of the data set in an attempt to discover information about the real-world process that has produced the data in the first place. This is achieved by exploiting the statistical information already present in the data set to rigorously infer causation relationships. The models built this way can then be used as predictors or as classifiers directly, or they can be studied to gain insights about the process that produced the data.

An essential causal model is the Bayesian network [2], which is a directed acyclic graph, the nodes of which represent the individual features of the data set, connected by edges which represent direct causal influences between the nodes. Apart from representing a feature, a node also contains the probability distribution of the feature conditioned on its immediate parents. By definition, the feature represented by the node is independent of all its ancestors, given its immediate parents [2]. It is important to note that having two unconnected nodes in a Bayesian network does not imply their independence. Instead, a more complex criterion named d-separation is employed to reveal the dependence status of two features in a Bayesian network [2].

Within a Bayesian network, the nodes immediately surrounding a node of interest form a structure named the Markov blanket. More specifically, the Markov blanket of a node is the subset of nodes containing the immediate parents, the immediate children, and the spouses of the node (the nodes that share its children). Figure 1 below shows a trivial example of a Bayesian network. By definition, the feature represented by the node is conditionally independent of the entire Bayesian network, given the features in its Markov blanket [2].

The complete Bayesian network is a rich source of causal information and having it available is highly desirable [3], but it is often unknown. However, one method of discovering it from the observed data set is to analyze the features to reveal their individual Markov blankets, then connect the blankets to each other [2]. The class of algorithms which can be used to build Bayesian networks from data in this manner are the Markov blanket discovery algorithms [4]. To our knowledge, most Markov blanket discovery algorithms found in literature are designed to perform a large number of statistical tests of conditional independence (CI) on the features of the given data set.

It must be noted that Markov blanket discovery algorithms function by discovering the Markov boundary (the smallest blanket) of features specifically, and not by discovering any possible Markov blanket—there can be multiple blankets for a node but only one boundary [2]. This is true in spite of the fact that the term ‘Markov blanket’ is used in other related articles [4,5,6]. However, this article will continue to use the term ‘Markov blanket’ and not ‘Markov boundary’, for consistency with its references. The Markov blanket discovery algorithms find themselves at an intersection point between causal inference and feature selection because they can be applied unmodified to solve problems in both fields [5,7], but interpreting their results to suit the perspective of the problem, either causal inference or feature selection.

While the algorithms have evolved and have become more sophisticated and efficient, requiring fewer tests of conditional independence, the CI tests themselves have remained a performance bottleneck due to their expensive computation and also due to the multiple passes of the data set required to compute a single one. A specific type of statistical independence test, the G-test, is used by multiple Markov blanket discovery algorithms, as described in Section 2. A theoretical overview of the G-test itself is provided in Section 4.

To address the CI test performance bottleneck, we have designed a computational optimization named ‘decomposed conditional mutual information’ (dcMI), which we propose and discuss in this article. Described briefly, dcMI involves writing the *G* statistic in the form of conditional mutual information, which can be further decomposed into a sum of easily reusable joint entropy terms. These terms are computed and cached once, but reused many times when computing subsequent G-tests during the same run of the algorithm or future runs on the same data set. Because it exploits the underlying connection between the G-test and the elements of information theory, our optimization is remarkably simple and highly effective. It is also widely applicable since it can be used by any algorithm which can be configured to use the G-test, while remaining strictly contained within the bounds of the G-test implementation itself. Moreover, no approximation is involved whatsoever: the *G* statistic has the exact same value when computed using our optimization and when computed in the canonical manner.

To demonstrate the efficiency gains produced by the dcMI optimization, a comparative experiment was performed using IPC–MB (Iterative Parent-Child Markov Blankets) [6], an efficient Markov blanket discovery algorithm. IPC–MB explicitly uses the G-test to determine conditional independence, applied sequentially over many permutations of the features in the data set. This is a characteristic inherited from the algorithms it builds on [4]. The experiment compares IPC–MB across multiple runs: with the G-test unoptimized, with the G-test enhanced with dcMI, and also with the G-test enhanced with an alternative optimization, AD–trees, as proposed by the original authors of IPC–MB themselves [6]. For each run of IPC–MB, the duration and memory consumption are measured. To achieve a comprehensive comparison, IPC–MB was configured to discover the Markov blanket of every feature of the data set, one at a time. The actual reconstruction of the Bayesian network from the Markov blankets was not performed because it is not relevant to the comparison.

As a secondary result, the comparative experiment revealed that computing the *G* statistic in its canonical (unoptimized) form consumes an average of 99.74% of the total running time of IPC–MB, even when applied on a small data set. This further emphasises the performance bottleneck inherent to CI tests and the necessity of their optimization.

This article is structured as follows: Section 2 presents the dcMI optimization in the context of existing Markov blanket discovery algorithms, while also describing other optimizations found in literature with which it shares features; Section 3 and Section 4 provide the theoretical background required to describe the dcMI optimization, namely the elements of information theory it relies on, and the statistical G-test, respectively; Section 5 describes the theoretical design of dcMI and its relation to the G-test, also including an example of how dcMI operates; Section 6 describes IPC–MB, the Markov blanket discovery algorithm used to demonstrate the efficiency of dcMI; Section 7 presents the comparative experiment in which dcMI is empirically evaluated using IPC–MB and compares its performance to alternative optimizations of IPC–MB found in literature; Section 8 summarizes the conclusions that emerge from the design of dcMI and from the experimental evaluation.

## 2. Related Work

The dcMI optimization was conceived, designed and implemented during research on Markov blanket discovery algorithms and their applications in causal inference and feature selection. For this reason, dcMI is discussed here only within this context, in spite of its generality. The current design of dcMI is rooted in previous efforts [8] on optimizing Koller and Sahami’s algorithm [9], the first algorithm to introduce Markov blanket discovery as an approach to feature selection. Although now obsolete, Koller and Sahami’s algorithm (KS) presented challenges interesting enough to merit further study, out of which an optimization named γ-decomposition emerged, the precursor to dcMI.

Algorithms that perform Markov blanket (MB) discovery have used the G-test extensively. However, the earlier algorithms of this family did not employ the G-test for testing conditional independence: Koller and Sahami’s (KS) algorithm [9], the first such algorithm, relied on a custom metric expressed as a Kullback–Leibler divergence (but later shown by Brown et al. [10] to be conditional mutual information); the IAMB algorithm [11] applied a threshold to the conditional mutual information of the tested features; the Grow-Shrink (GS) algorithm [3] used the canonical $\chi^{2}$ test.

Leaving the early algorithms aside, the PC algorithm [12] is to our knowledge the first MB discovery algorithm to employ the G-test to determine conditional independence, although the earlier GS algorithm was the first algorithm to be proven correct [4]. Inspired by the aforementioned IAMB algorithm and attempting to attain better efficiency, the MMMB [7] and HITON [13] algorithms were published. Both followed the same principles as IAMB, while also using the G-test for conditional independence. Pena et al. [5] later showed that the proofs of MMMB and HITON are incorrect and propose the PCMB algorithm, which builds upon IAMB, MMMB, and HITON and which also uses the G-test. To our knowledge, the next algorithm of this lineage is IPC–MB [6], which improves on the design of PCMB to further reduce the number of required CI tests. The IPC–MB algorithm uses the G-test as well, following in the footsteps of PCMB and its predecessors. IPC–MB is the Markov blanket discovery algorithm chosen for this article to showcase the dcMI optimization.

At a closer look, the aforementioned algorithms apply the G-test as it is described by Spirtes and Scheines in ‘Causation, Prediction and Search’ [12], including the method of computing the degrees of freedom of a test, as well as the minimum amount of samples required to accept the result of a test.

To increase the computational efficiency of Markov blanket discovery algorithms, specialized data structures such as AD–trees can be used, as proposed by the authors of IPC–MB themselves [6]. An AD–tree is a data structure that allows for very fast retrieval of sample counts that match a query. AD–trees were proposed by Moore and Lee [14] as a very general solution to accelerate a wide range of machine learning problems. The IPC–MB++ algorithm [6] uses such a pre-built static AD–tree to successfully overcome the CI test bottleneck but at great cost, as later discussed in Section 7.

To address the shortcomings of static AD–trees, Komarek and Moore [15] proposed dynamic AD–trees. Dynamic AD–trees are built while running the algorithm, not before. New information is added at runtime only as a result of actual queries, making the tress much smaller and more efficient. This mode of operation makes dynamic AD–trees far more similar and comparable to the dcMI optimization than the static ones.

Furthermore, an alternative to dynamic AD–trees are AD+trees, proposed by Moraleda and Miller [16] to address the inefficiencies of dynamic AD–trees when dealing with stochastic queries, as opposed to more systematic queries, while also being more compact. However, since IPC–MB++ performs queries quite systematically [6] (a characteristic shared by most algorithms in its family), AD+trees were not implemented for IPC–MB++ in the experimental comparison presented in this article. Instead, the experiment is focused only on the aforementioned static and dynamic AD–trees.

Another general approach for accelerating CI tests is the PFBP algorithm proposed by Tsamardinos et al. [17]. PFBP attempts to improve the time efficiency of CI tests by partitioning the data set into Data Blocks, which are sent to separate computational nodes (i.e., separate machines) in order to be processed in parallel. Each computational node performs CI tests locally, on their received Data Block, in order to produce local *p*-values. The local *p*-values are later centralized to a main node, which eventually produces combined global *p*-values using Fisher’s combined probability test [17]. However, the local *p*-values cannot be used to reconstruct the exact value of the global *p*-value. Because the PFBP is intended to be a statistical feature filter and not a Markov blanket discovery algorithm, the global *p*-values are used directly for ranking features among themselves in each iteration and for making decisions on selecting or rejecting them. According to the authors, the PFBP algorithm allows the use of any type of conditional independence test to generate the local *p*-values. Inspired by their work, a method of distributing the computation of the joint entropy terms used in dcMI is currently being designed. Such a method will rely on the fact that each term can be computed independently of any other. Moreover, it is likely that a parallel dcMI implementation will require an algorithm with different characteristics, replacing IPC–MB.

To our knowledge, there are no implementations of the G-test that attempt to optimize it by exploiting its relationship to information theory. Still, an alternative method of computing the G-test was presented by Tsagris [18]. According to the article, it is faster to build a log-linear model on the data set using Poisson regression and compute its deviance than to compute the G-test directly. This relies on the fact that the deviance of such a model also has $\chi^{2}$ as its limiting distribution [19], just like the *G* statistic [20]. The method presented by Tsagris [18] has not been replicated in the comparative experiment presented in this article; therefore, its efficiency has not been compared to the dcMI optimization.

Apart from the G-test, there are other methods for testing independence that rely on mutual information. Of particular interest is the work of Al-Labadi et al.[21], who propose a test of independence for continuous features that relies on computing mutual information using Bayesian nonparametric estimators for entropy. In theory, this would allow Markov boundary discovery algorithms to operate on data sets with continuous variables.

## 3. Notes on Information Theory

This section introduces the fundamental elements of information theory required to describe the dcMI optimization. The elements of interest are entropy, joint entropy, mutual information, and conditional mutual information. Considering the abstract character of the concepts presented below, the more general term ‘variable’ is used instead of ‘feature’, although, for the purposes of this article, these two terms are interchangeable.

Information theory quantifies the informational content of the probability distribution of a variable *X* using a measure named entropy [22], defined as:$$H\left(X\right)=-\sum_{x\in V_{X}}P\left(x\right)\cdot \log\left(P(x)\right)$$
The multivariate extension of entropy is named joint entropy [22]. Considering variables $X_{1}$, $X_{2}$, *…*, $X_{m}$, their joint entropy is defined as follows:$$H\left(X_{1},X_{2},\ldots ,X_{m}\right)=-\sum_{x_{1}\in V_{X_{1}}}\ldots \sum_{x_{m}\in V_{X_{m}}}P\left(x_{1},x_{2},\ldots ,x_{m}\right)\cdot \log\left(P(x_{1},x_{2},\ldots ,x_{m})\right)$$
or, if $X_{1},X_{2},\ldots ,X_{m}$ is noted simply with X (the bold letter denoting a set of variables, instead of a single one), their joint entropy can be written compactly:(1)$$H\left(X\right)=-\sum_{x\in V_{X}}P\left(x\right)\cdot \log\left(P(x)\right)$$

Joint entropy has the very convenient property of not depending on the order of its arguments, a property which will be intensively exploited by the dcMI optimization, as will be described in Section 5.

Entropy also has a conditional form:$$H\left(X|Y\right)=-\sum_{x\in V_{X}}\sum_{y\in V_{Y}}P\left(x,y\right)\cdot \log\left(P(x|y)\right)$$

The difference between the entropy of *X* and its conditional entropy given *Y* is named mutual information [22], defined as:I(X;Y)=H(X)−H(X∣Y)

Mutual information is interpreted as the amount of information shared by *X* and *Y*. It is also symmetric in its arguments, meaning that I(X;Y)=I(Y;X). There is a conditional form of mutual information as well [22]. Let *Z* be a third variable:(2)I(X;Y∣Z)=I(X;Y,Z)−I(X;Z)
The conditional mutual information of *X* and *Y* given *Z* is interpreted as the amount of information shared by *X* and *Y* that still cannot be explained away by knowing *Z*.

Of notable importance is the fact that conditional mutual information can be decomposed into a sum of joint entropy terms:(3)$$\begin{array}{cc} I(X\;;Y\;|Z) & =I(X\;;Y,Z)-I(X\;;Z)\\ & =H\left(X,Z\right)+H\left(Y,Z\right)-H\left(X,Y,Z\right)-H\left(Z\right) \end{array}$$
As will be discussed later, this decomposition features prominently in the proposed dcMI optimization and has determined the choice of its name.

## 4. The G-Test

The G-test is a statistical hypothesis test with the formal name of log-likelihood ratio test [23]. Therefore, it is a tool used to determine which of two competing hypotheses is more likely to be true after empirically observing a series of events. More specifically, it can be applied in the context of a data set in order to determine whether it is more likely that two features are independent (the null hypothesis, $H_{0}$) or, on the contrary, if they are dependent (the alternative hypothesis, $H_{1}$). For brevity, this article will keep using the name ‘G-test’ for the test itself and *G* for its statistic, even though ‘log-likelihood ratio test’ is its canonical name. The name $G^{2}$ also appears in literature [24].

Although the G-test can be used as a generic hypothesis test, this article focuses on applying the G-test as a test of independence between two multinomial features *X* and *Y*, after experimentally observing them and recording *N* instances. To test the unconditional independence of *X* and *Y*, let the null hypothesis $H_{0}$ be X⊥⊥Y, thus expressing the independence of *X* and *Y*, and let $H_{1}$ be the alternative hypothesis asserting the opposite, X⊥⊥/   Y, thus expressing a dependence relationship between *X* and *Y*.

After experimentally obtaining instances of *X* and *Y*, the test of independence becomes a problem of determining how likely or unlikely $H_{0}$ is in light of this evidence. The *G* statistic quantifies the likelihood of $H_{0}$ versus $H_{1}$, and is defined as follows:(4)$$G\left(X,Y\right)=2N\sum_{x\in V_{X}}\sum_{y\in V_{Y}}P\left(x,y\right)\ln\frac{P(x,y)}{P(x)\cdot P(y)}$$

Here, $V_{X}$ and $V_{Y}$ are the sets of values that can be taken by the *X* and *Y* features. In addition, *x* and *y* are elements of these sets, respectively.

After computing the *G* statistic, one can determine how likely it is that *X* and *Y* are indeed independent, given the evidence. This is done by evaluating the probability of obtaining the value of *G* yielded by Equation (Equation 4) under its limiting probability distribution, $\chi^{2}$ [20]. This results in a *p*-value, a real number in the interval [0,1 which is the conditional probability of observations as extreme as expressed by *G* given that $H_{0}$ is true. The lower the *p*-value, the less probable the observations are under $H_{0}$; therefore, for low *p*-values, the $H_{0}$ may be rejected, and $H_{1}$ may be considered true instead.

It is up to the researcher to choose a threshold α for *p*-values under which $H_{0}$ is rejected and *X* and *Y* are considered dependent. The value α is formally known as the significance of the test.

It is important to note that the $\chi^{2}$ distribution is parametric, requiring a number of degrees of freedom (DoF) to be specified [24] before computing the *p*-value of the *G* statistic. This distribution parameter is required due to the definition of the $\chi^{2}$ distribution: it is the sum of the squares of a number of DoF independent standard normal variables [24]. The value of DoF must be calculated independently for each instance of the test. For consistency with its references, the DoF values throughout this article will be calculated in a manner similar to the one described by Spirtes et al. [12]. A more detailed description of how this calculation is performed is provided in Appendix E.

Apart from testing the marginal independence of two features, the G-test can also be used to test conditional independence as well, which is of central importance in Markov blanket discovery. If an experiment has recorded more features than just *X* and *Y*, one can then perform tests of conditional independence between *X* and *Y*, given some subset of the other recorded features, in order to investigate any causality relationships among them [2]. Let Z be such a conditioning set, with the restrictions X∉Z and Y∉Z. The *G* statistic for a conditional test of independence can then be calculated as:(5)$$G\left(X,Y|Z\right)=2N\sum_{x\in V_{X}}\sum_{y\in V_{Y}}\sum_{z\in V_{Z}}P\left(x,y,z\right)\ln\frac{P(x,y|z)}{P(x|z)\cdot P(y|z)}$$

## 5. The G Statistic and dcMI

This section contains the discussion on the theoretical relationship of the G-test with information theory and how it is exploited by dcMI, our proposed optimization.

As stated in the introduction, a significant performance bottleneck manifests itself when an algorithm must compute a large number of tests of conditional independence (CI). This is the case of Markov blanket discovery algorithms such as IPC–MB. These algorithms apply the G-test on various permutations of the features in a data set in order to discover relationships of conditional independence.

The probabilistic form of the *G* statistic shown in Equation (Equation 5) is visibly similar to the definition of the conditional mutual information presented in Equation (Equation 2). Consequently, Equation (Equation 6) compactly represents the *G* statistic as conditional mutual information:(6)G(X,Y∣Z)=2N·I(X;Y∣Z)
Moreover, using the decomposition of conditional mutual information from Equation (Equation 3), the *G* statistic can be written using a simple sum of joint entropy terms:(7)$$G\left(X,Y|Z\right)=2N\cdot \left[H\left(X,Z\right)+H\left(Y,Z\right)-H\left(X,Y,Z\right)-H\left(Z\right)\right]$$
Equation (Equation 7) is the form of the *G* statistic exploited by the dcMI optimization. This decomposed form is highly desirable when applying the G-test for conditional or unconditional independence, especially for Markov blanket discovery. When calculating the *G* statistic for various permutations of the features *X*, *Y*, and those in the set Z, it is highly likely that some or all of the joint entropy terms of the decomposition will be reused directly from previous calculations of *G*. This reusal is a direct consequence of the indifference of joint entropy to the order of its arguments. Moreover, note that terms may be reused between two separate calculations of *G* even when only some of the features on which they are applied are shared. In effect, the dcMI optimization stores and reuses these joint entropy terms for each computation of the *G* statistic that takes place during the run of an algorithm. The only requirement is to compute *G* in its decomposed form as in Equation (Equation 7).

It can be shown that reusing joint entropy terms when computing G-tests requires $\frac{M(M-1)}{2}$ times fewer probability distributions to be computed from a data set of *M* features in the combinatorically exhaustive case (all possible G-tests performed on *M* features). The proof is provided in Appendix A.

The dcMI optimization stores joint entropy terms in a cache named Joint Entropy Table (JHT). There is no specific requirement for the underlying data structure of the JHT: any associative mapping structure will suffice, as long as it can store and retrieve real numbers (values of joint entropy), indexed by unique sets of feature indices. Retrieving the joint entropy of a set of features from the JHT makes it unnecessary to access the data set for further information.

There is a drawback to caching terms and avoiding the data set entirely. As stated in Section 4, computing a G-test also requires specifying the number of degrees of freedom (DoF) of the $\chi^{2}$ distribution [12]. The DoF must be recalculated for each individual test based on the features involved. However, when the JHT already contains the joint entropies of the tested features, accessing the data set will still be required, only to compute the DoF. This would negate the advantage brought by the JHT. For the purposes of this article, comparability with DoF computation as implemented in previous work must be preserved; therefore, the dcMI optimization may not use a different DoF computation method than the ones used by the other G-test configurations in this experiment. To mitigate this drawback, a secondary cache is maintained for DoF values, named DoF cache, again indexed by unique sets of feature indices. Maintaining the DoF cache is more complicated than maintaining the JHT itself and consumes more memory. The motivation for maintaining the DoF cache and its operation is described in Section 6.2. Future work will attempt to provide a better approach, obviating the need for the expensive DoF cache.

To better illustrate how joint entropy terms are reused by G-tests performed on permutations of the same features, a simple and intuitive step-by-step example is provided in Appendix B.

## 6. The Iterative Parent-Child Markov Blanket Algorithm

The Iterative Parent-Child Markov Blanket algorithm, IPC–MB for short, is an efficient Markov blanket discovery algorithm [6]. As described in Section 2, IPC–MB is one of the algorithms in its class which explicitly uses the G-test to establish conditional independence (CI) and to discover the local structure around a target feature within a Bayesian network. It guarantees finding the Markov blanket of a target feature, and it is proven correct by its authors. To our knowledge, it is one of the most efficient algorithms in its class with respect to the number of CI tests it performs.

Just like PCMB [5], IPC–MB divides the problem of discovering the Markov blanket of a target feature *T* in two subproblems: (1) the discovery of its parents and children PC(T), and (2) the discovery of its spouses, which are the nodes that share its children SP(T). The Markov blanket is the union of these two sets, thus it is defined as MB(T)=PC(T)∪SP(T). Due to the inherent structure of a Bayesian network, the set SP(T) can be found by iterating over the features X∈PC(T), then finding PC(X) for each *X* and testing if each element of each PC(X) is conditionally independent of *T* given PC(T). If an element of any PC(X) is found to be not independent by this test, then it belongs to SP(T).

The improvement brought by IPC–MB over PCMB is the order in which tests of conditional independence are performed. While PCMB does not mention a specific order, IPC–MB ensures that CI tests are performed first conditioning on the small subsets of features, then on the larger ones. Therefore, it first exhausts the tests of unconditional independence (size 0), then the tests of conditional independence given the subsets of size 1, then 2 and so on [6]. Conditioning on incrementally increasing subsets of features has the advantage of avoiding expensive and likely unnecessary tests that condition on larger subsets.

It is important to note that IPC–MB neither has nor requires direct access to the actual features and samples of the data set. Instead, all the information needed by IPC–MB is provided by the G-test as a black box in the form of true/false answers concerning the conditional independence of two specific features given a subset of features. Moreover, the G-test is queried using only feature indices, which means it is the responsibility of the G-test implementation to acquire the information it needs regardless of whether this involves consulting the data set directly or extracting information from other data structures. Briefly put, IPC–MB does not need to know how the G-test is computed nor how the information required to compute the test is acquired.

Upon completion, IPC–MB returns a set of indices which represent the features in the data set that form the Markov blanket of the target feature *T*.

### 6.1. IPC–MB and AD–Trees

The authors of IPC–MB also proposed an improved variant of IPC–MB named IPC–MB++ [6], which changes how sample counts are retrieved from the data set and fed into the G-test computation. By default, the unoptimized G-test iterates over the samples of the data set and counts those that match specific values in order to build the probability mass functions it requires. In contrast with the default IPC–MB behavior, IPC–MB++ replaces sample counting for G-tests with queries into an AD–tree [14] with the intention of gaining efficiency.

An AD–tree is a data structure built to contain the sample counts of all possible combinations of instance values for any subset of features. As its name indicates, it is a tree and it contains two types of nodes named AD–nodes (sample count nodes) and Vary–nodes (feature nodes). The root of the tree is an AD–node containing the count of all samples; therefore, it contains the answer to the most universal sample count query. The children of the root are Vary–nodes, one for each feature of the data set. The immediate children of every Vary–node are also AD–nodes, one for each possible instance value that the feature represented by the parent Vary–node can take. Every such AD–node is the root of a new subtree, which can itself be queried for sample counts, with the restriction that the query matches its corresponding instance value for the feature represented by its parent Vary–node. For example, the number of samples for which $X_{1}=8$ and $X_{2}=16$ can be found by starting at the root, then visiting the Vary–node corresponding to $X_{1}$, then its AD–node child corresponding to the value 8. This AD–node will also have Vary–node children for all features (except for $X_{1}$ itself); therefore, the Vary–node corresponding to $X_{2}$ must be visited next. The final answer is the sample count stored in its AD–node child corresponding to the value 16. See Appendix C for a simple example of building a dynamic AD–tree. Note that no AD–nodes are generated for sample counts of 0, to save memory. When a query of the AD–tree finds no corresponding node, it implicitly returns 0.

When building AD–trees, an ordering of the features must be first established. The AD–tree will follow this ordering when expanding. It is essential to always sort the features in a query by the same ordering as the one used when the AD–tree was built.

If the AD–tree has been built in its entirety before it was ever queried, it is a static AD–tree [14]. By contrast, if the AD–tree is empty at the time of the first query and will be expanded on-demand while it is queried, it is a dynamic AD–tree [15]. The static AD–tree has the obvious advantage of readily providing the answers to sample count queries, while the dynamic AD–tree will still require access the data set and count samples if the subtrees required by the query have not been expanded yet. However, the static AD–tree has the severe disadvantage of non-trivial memory consumption, as well as being expensive to build in the first place, while the dynamic AD–tree does not expand more than the queries require it to expand, saving both time and memory.

Deciding whether to employ a static AD–tree or a dynamic one is a matter of compromise. Some applications may benefit from the low latency of the static AD–tree, while others may benefit from the low resource consumption of the dynamic AD–tree. Moreover, the static AD–tree completely obviates the need to keep the data set available because it already contains all the information in the data set. By contrast, the dynamic AD–tree will still need access to the data set when it is queried because it might need to expand new nodes.

To reduce the memory consumption of an AD–tree, its authors [14] also propose two pruning-like optimizations:eliding AD–nodes for the Most Common Value (MCV) whenever creating the children of a Vary–node; MCV elision is a significant optimization for AD-trees because it prevents the expansion of the subtree under the AD–node with the highest sample count among the children of a Vary–node. This optimization relies on the assumption that a higher sample count would require a subtree with more nodes to be generated. This is because it is more probable to encounter a greater variety of samples in a larger subset of the samples, and a greater variety of samples implies that AD–nodes will often contain positive (non-zero) sample counts, even at greater depths of the subtree. To recover the information elided by an unexpanded subtree, the querying process must reconstruct missing sample counts at runtime, a more complex procedure;replacing as many small subtrees as possible at the deeper levels with Leaf-List nodes; Leaf-List nodes prevent the expansion of subtrees when their total sample count drops under a manually given threshold but they keep references to (or copies of) the original samples in the data set. This threshold is named the Leaf-List Threshold (LLT). In case a query needs to descend into a subtree replaced by a Leaf-List node, it will count the referenced raw samples instead, directly from the data set.

Employing these optimizations increases the logical complexity of querying because the elided information must be inferred at runtime from other information existing in the AD–tree [14]. These optimizations are applicable to both static and dynamic AD–trees [15]. See Appendix D for more details on these optimizations.

Computing a G-test requires the conditional probability mass function (CPMF) of a subset of features. This means that the raw sample counts stored in the AD–trees must be first assembled into a CPMF before computing the *G* statistic and computing the number of degrees of freedom required for the CI test, as described in Appendix E. While not logically straightforward, assembling a CPMF from an AD–tree suits the tree structure well because the design of the AD–tree makes recursive querying possible.

When IPC–MB++ was proposed, it employed a static AD–tree [6] as the source of sample counts when computing G-tests. However, in spite of the pronounced differences in behaviour between static and dynamic AD–trees, they are essentially the same data structure and are equivalent from the perspective of a consumer algorithm. As a consequence, IPC–MB++ is agnostic of the type of AD–tree it queries. This fact makes it possible to configure an implementation of IPC–MB++ to use either a static or dynamic AD–tree with no further modification whatsoever.

### 6.2. IPC–MB and dcMI

The performance of IPC–MB can also be improved using the dcMI optimization. Instead of computing the *G* statistic in its usual form, it is computed as a scaled sum of joint entropy terms, as described in Section 5. This will exploit the abundant reusal of joint entropy terms between *G* statistic computations for various permutations of features and thus will greatly reduce the total running time of the algorithm. The dcMI optimization requires no parameters to be set.

The design of the dcMI optimization is straightforward: the computation of the *G* statistic is done by summing joint entropy terms stored in the Joint Entropy Table (JHT), corresponding to the features currently being tested for conditional independence (CI) and to the features in the conditioning set. If a CI test requires a joint entropy term that is not currently stored in the JHT, then it must be calculated from the data set and added to the JHT as follows:the features required for the missing joint entropy term are retrieved from the data set;their joint probability mass function is computed;their joint entropy is computed;the joint entropy is stored in the JHT under a key that uniquely identifies the specific subset of features; andthe new joint entropy value is now available to compute the *G* statistic.

These steps are performed as many times as required, on demand, a maximum of 4 times per CI test, as per Equation (Equation 7).

At the beginning, the JHT is empty and joint entropy terms are added as IPC–MB performs CI tests. Moreover, the JHT may be shared between IPC–MB runs performed on the same data set, accelerating the algorithm further.

An essential piece of information required to perform a G-test is the number of degrees of freedom (DoF) of the limiting $\chi^{2}$ distribution corresponding to the features being tested. In the form in which it was proposed, IPC–MB computes the number of DoF as proposed by Spirtes et al. [12]. This specific method requires the knowledge of zero values in the joint probability mass function of the features in the test. A method that is more computationally expensive but more mathematically appropriate is to employ linear algebra, in effect treating the joint probability mass function as a system of linear equations and finding its number of free variables. An overview of this approach is discussed by Lamont [25]. In this specific implementation, the number of DoF is not computed exactly as suggested by Spirtes et al. [12], but in a slightly different way. The method is described in Appendix E.

A straightforward implementation of DoF computation would have put the dcMI optimization at a significant disadvantage: having to eliminate the zero values from P(X∣Z), P(Y∣Z) and P(Z) for each CI test would have required counting samples directly from the data set. This would have negated the entire efficiency gain brought by caching joint entropy terms. On the contrary, AD–trees do not have this disadvantage because they already contain the sample counts, which have to be assembled into probability mass functions anyway in order to compute the unoptimized *G* statistic.

To prevent having to compute the joint probability mass functions of the tested features in each CI test, an alternative approach was employed. Whenever a new JHT entry is added and a probability mass function is indeed built directly from the data set, a secondary cache is updated as well. This is the DoF cache, which contains the degrees of freedom computed for all the possible permutations of the encountered subsets of features. More specifically, when IPC–MB requests a CI test between the features $X_{i}$ and $X_{j}$ conditioned on $X_{k}$, then the DoF cache is immediately updated with the degrees of freedom for the tests between $X_{i}$ and $X_{k}$ conditioned on $X_{j}$ and between $X_{j}$ and $X_{k}$ conditioned on $X_{i}$, even if the CI tests which would require the latter two entries have not been encountered yet. This is the only opportunity available to compute the degrees of freedom for the aforementioned permutations because the data set will not be accessed later for the joint probability mass function of $X_{i}$, $X_{j}$ and $X_{k}$.

Using this approach, IPC–MB with the dcMI optimization will be able to compute DoF in the same way as IPC–MB++ does using the AD–trees, but at the cost of managing and storing an extra cache which contains precomputed numbers of DoF. It is likely that many of the entries of the DoF cache are never used, but this aspect has not been measured in the experiment.

The DoF cache is only required in order to ensure that the G-tests computed with dcMI are identical with the ones computed from AD–trees or those computed without optimization, according to the method proposed by Spirtes et al. [12]. If this specific restriction is lifted, different methods for DoF computation may be used. A discussion on alternative DoF computation methods is beyond the scope of this article, but it is essential and will be researched further in the future.

It is also possible to implement eviction mechanisms for the JHT and the DoF cache. Such mechanisms would routinely remove entries from the JHT and DoF cache, thus maintaining a lower memory consumption and enabling IPC–MB to run on far larger data sets than the ones in the comparative experiment presented here. However, this experiment was restricted to data sets that did not require eviction.

## 7. A Comparative Experiment

To evaluate the dcMI optimization empirically, a comparative experiment was performed on two multinomial data sets (ALARM and ANDES, described in Section 7.2) by running the IPC–MB algorithm in four configurations. The four configurations are the following:*GDefault*, using an unoptimized G-test implementation, which had to count raw samples in the data set for calculating each CI test; this is the default configuration of IPC–MB as proposed by Fu and Desmarais [6]; this configuration is only provided as reference, where feasible;*GStADt*, using a G-test implementation which retrieves information from a pre-built static AD–tree instead of reading the data set directly (except when Leaf-List nodes are built, as described in Section 6.1); this configuration cannot be run unless the static AD–tree has been already built on the chosen data set; this configuration allows the variation of the LLT parameter (see Section 6.1); this configuration is only provided as reference, where feasible;*GDyADt*, using a G-test implementation which retrieves information from a dynamic AD–tree, instead of reading the data set directly; this configuration requires access to the data set, but it does not require a tree-building phase and can start immediately; this configuration allows the variation of the LLT parameter;*GdcMI*, using a G-test implementation which relies on dcMI to calculate CI tests, which means that it maintains a Joint Entropy Table (JHT) and a DoF cache, both described in Section 5, which are consulted to compute the *G* statistic without accessing the data set at all for most CI tests.

As stated in Section 2, the dcMI optimization is most comparable to the dynamic AD–trees, due to its mode of operation. For this reason, the focus of this experiment is to compare these two optimizations. However, the experiment also contains the unoptimized IPC–MB and static AD–trees in order to provide a more complete overview. Note that, regardless of the G-test configuration, there is a single method of computing the number of degrees of freedom for a G-test, described in Appendix E, used throughout the experiment.

The experiment consists of applying IPC–MB in each configuration to discover the Markov blanket (MB) of every feature in the data sets ALARM and ANDES. While the discovery of the MB of a single feature under a specific configuration constitutes a single run of IPC–MB, the experimental unit actually consists of as many runs with a configuration as the data set has features. The experiment records the time required to complete each of the IPC–MB runs, as well as the memory consumed by the data structures employed by the configurations, namely the AD–trees, the JHT and the DoF cache. In the context of this experiment, these data structures were collectively named the ‘optimization structures’.

The LLT parameter of the configurations based on AD–trees was restricted to the values 0, 5, and 10. However, these values do not represent the absolute numbers of samples below which an AD–node would not be expanded further. Instead, these values are percentages of the total samples of the data set.

Implementing AD–trees efficiently took considerable effort due to the overall complexity of the querying logic. While a straightforward but correct implementation was expected to be inefficient, the amount of work required to produce an implementation of acceptable efficiency accounted for the vast majority of the total implementation effort expended for this experiment. Profiling had to be used extensively to bring the AD–tree implementations (both static and dynamic) to an efficiency that made the comparison to dcMI meaningful, both in running time and memory, even when accounting for the overhead introduced by DoF caching.

A core aspect of the experiment is that all four configurations produce exactly the same results. This fact is a consequence of the design of each configuration: the AD–trees contain the same counts which would have been retrieved from the data set directly, while dcMI is only an equivalent writing of the definition of the *G* statistic. For this reason, no measurement of accuracy was performed and the comparison between the IPC–MB configurations is only made regarding time and memory.

For all four configurations, the significance of the G-test was set to 0.05. This means that, for all G-tests, a *p*-value below 0.05 will result in the tested variables being considered dependent, while a *p*-value above 0.05 will result in the variables being considered independent. No dynamic per-test adjustments are made to the significance.

The experiment is focused on the most intensive use-case of the IPC–MB algorithm, namely to find the Markov blankets of all the features in each data set. This exhaustive use-case emulates how IPC–MB would be applied for reconstructing the entire Bayesian network of the data sets, as opposed to the simpler use-case of applying IPC–MB to find the MB of a single feature. For this reason, the optimization structures employed by the algorithm configurations (AD–trees, JHT, DoF caches) were shared between all algorithm runs of the same configuration and on the same data set, as follows:the *GStADt* configuration shares the static AD–trees among its runs on each specific data set given an LLT argument, in order to amortize the cost of building it once per data set [6];the *GDyADt* configuration shares the dynamic AD–trees among its runs on each specific data set given an LLT argument, initially empty, but which are expanded as IPC–MB performs G-tests on the features of the data sets;the *GdcMI* configuration shares the JHT instances and DoF caches among its runs on each specific data set.

The IPC–MB runs were grouped into two subexperiments corresponding to the data sets they were run on, ALARM and ANDES.

The entire experiment was performed on a modern machine of average specifications: Intel Core i3-8100T processor with 16 GB of DDR4 RAM and an SSD drive for storage, running an up-to-date installation of Fedora 32. To make sure the resources of the machine were dedicated entirely to the experiment, a preparation step was run before starting the experiment which included closing the desktop environment, setting the operating system in multi-user mode only and shutting down all the networking subsystems, as well as other userspace processes (such as dbus, pulseaudio, cups).

### 7.1. Implementation

The entire source code of the experiment is freely available under the terms of the GPLv3 license and was published on the Internet as a Git repository [26]. This repository also contains a miniature version of the entire experiment, intended as a short demonstration only.

The entire experimental framework, the IPC–MB algorithm, the G-test configurations and the optimization structures were all implemented in Python 3 without relying on other implementations. This lack of dependencies is an unfortunate fact caused by the lack of acceptable public implementations of the IPC–MB algorithm and AD–trees. This means that the IPC–MB and IPC–MB++ implementations used in this experiment are, to our current knowledge, the only publicly available ones.

However, the experiment does rely on the open-source Python packages numpy [27], scipy [28], and larkparser [29].

The implementation of the IPC–MB algorithm was validated by automatic tests involving the d-separation criterion applied on the original Bayesian network as the CI test. Because of this, the efficiency of IPC–MB itself could be compared to other algorithms in its class, with respect to the number of CI tests required to find a Markov blanket. The implementations of the four configurations of G-tests are validated by automatic tests which ensure that all of them reach the same value of the *G* statistic and the same number of degrees of freedom.

The AD–tree was implemented as a general-purpose data structure. The implementation is intended to be usable in external projects without the rest of the experimental framework. Both the static and dynamic forms are available.

### 7.2. Data Sets

The data sets used in the experiment were synthetically generated from two publicly available Bayesian networks, retrieved as BIF files (Bayesian Interchange Format) from the bnlearn R package [30]. A BIF reader was implemented using the larkparser Python package [29] to load them into the experimental framework. To generate the actual samples for the data sets, a custom-built Bayesian network sampler was implemented, which produced randomly generated samples on-demand, respecting the conditional dependence relationships between the features. This sampler provides the values of all the features of a sample, thus generating full sample vectors. Similar sample generators were found as libraries, but they lacked important features, such as generating a sample containing the values of all features, not only the value of a target feature.

The ALARM Bayesian network [31] was produced by the ALARM diagnostic application, which used the network to calculate probabilities that aid in differential diagnosis and patient monitoring. This Bayesian network was categorized as a ‘medium network’ by the bnlearn package. The ANDES Bayesian network [32] is a model used to assist in long-term student performance collected by the ANDES tutoring system. This network was categorized as ‘very large’ by the bnlearn package.

The following subsections describe the two subexperiments performed on these data sets.

#### 7.2.1. The ALARM Subexperiment

The ALARM data set contains 37 features; therefore, this subexperiment contains the following runs of IPC–MB:37 runs of GDefault111 runs of GStADt, further subdivided into 3 groups of 37 runs, one for each of the selected values for the LLT parameter, namely 0, 5, and 10; the static AD–trees are shared only among the runs in each LLT group;111 runs of GDyADt, further subdivided into 3 groups of 37 runs, one group for each of the selected values for LLT, namely 0, 5, and 10; the dynamic AD–trees are shared only among the runs in each LLT group;37 runs of GdcMI.

The ALARM data set was instantiated in three sizes: 4000 samples, 8000 samples, and 16,000 samples. The ALARM subexperiment was therefore performed in its entirety three times, once for each instance of the data set. In total, 888 runs of IPC–MB were performed in this subexperiment.

Because the AD–trees cannot be shared among data set sizes, nor among configurations with different values for LLT, a total of 18 AD–trees were built: nine static AD–trees and nine dynamic AD–trees. The dynamic AD–trees do not require a building step because they are built at runtime. However, static AD–trees must be pre-built. The durations required to build the nine required static AD–trees are shown in Table 1.

Table 2 presents the measurements recorded during the ALARM subexperiment. The columns mean the following:‘Time’: the total time spent by IPC–MB performing only CI tests; these durations include neither initialization time, nor the time IPC–MB needs to perform its own computation; the time to build the static AD–trees is excluded as well;‘Rate’: the average number of CI tests performed per second;‘Mem’: the total amount of memory consumed by the optimization structures, in megabytes; for AD–tree–based configurations, this is the size of the AD–tree alone; for GdcMI, it is the combined size of the JHT and DoF cache.

The results of the ALARM subexperiment, shown in Table 2, provide an overview of the efficiency of the four configurations. The GDefault configuration, which computes the *G* statistic without any optimization, is the slowest configuration, unsurprisingly. The configurations GStADt and GDyADt performed similarly to each other for each of the LLT values, but GDyADt has the significant advantage of constructing a very small dynamic AD–tree, even for LLT 0 (no leaf-list elision).

For emphasis, the static AD–tree built for GStADt with LLT 0 is around 2.5 GB in size, while the GDyADt with LLT 0 configuration produced an AD–tree of merely 49.6 MB. In spite of the considerable difference in memory consumption, they finished discovering all the Markov blankets in the ALARM data set within 1 or 2 s of each other (depending on the number of samples in the data set instance).

Containing our proposed optimization, the GdcMI configuration outperforms all the others on all three data set instances due to the high hit rates in the JHT, recorded at 97.0%, 97.5%, and 97.5% for the 4000, 8000, and 16,000 samples, respectively. On the data set instances of 4000 and 8000 samples, GdcMI discovers the 37 Markov blankets in less than half the time required by the next fastest configuration, GStADt with LLT 0, while consuming under 5 MB of memory. In the case of the 16,000 sample data set instance, the GdcMI configuration is still the fastest, but the difference to the next fastest configuration is not as dramatic mostly due to the fact that the joint entropy terms were still expensive to compute and also due to the low number of features in the data set, which caused only modest reusal of the terms in the JHT. However, the GdcMI configuration required 8.6 times less memory than the most memory-efficient AD–tree configuration, GDyAdt with LLT 10, which ranked third by time.

#### 7.2.2. The ANDES Subexperiment

The ANDES data set contains 223 features, six times as many as the ALARM data set. For this reason, the configuration GDefault was not even attempted on ANDES, but also due to its low CI test rate in the ALARM subexperiment. Moreover, attempts to build the static AD–trees for the GStADt configurations were aborted by the OS kernel due to their excessive consumption of memory. As a consequence, the ANDES subexperiment only contains the GDyADt and GdcMI configurations. As stated earlier, the proposed dcMI optimization is most comparable to the dynamic AD–tree anyway. To summarize, the ANDES subexperiment contains the following runs:669 runs of GDyADt, further subdivided into three groups of 223 runs, one group for each of the selected values for LLT, namely 0, 5, and 10; the dynamic AD–trees are shared only among the runs in each LLT group;223 runs of GdcMI.

Like the ALARM data set, the ANDES data set was also instantiated in three sizes: 4000 samples, 8000 samples, and 16,000 samples. The ANDES subexperiment was therefore performed in its entirety three times, once for each instance of the data set. In total, 892 runs of IPC–MB were performed in this subexperiment.

Because the AD–trees cannot be shared among data set sizes nor among configurations with different values for LLT, a total of nine dynamic AD–trees were built at runtime. No static AD–trees were built, as they were unfeasible on the experiment machine.

Table 3 presents the measurements recorded during the ANDES subexperiment. The table is missing results for the GDefault and GStADt configurations, due to their unfeasibility on the ANDES data set, as mentioned earlier. The remaining configurations, GDyADt and GdcMI, show a much more dramatic difference in efficiency on the ANDES data set. The columns have the same meaning they had for the ALARM subexperiment.

Firstly, the GDyADt configurations are slowed down significantly by deep-level queries into the dynamic AD–tree caused by the higher number of features involved in CI tests. This requires larger probability mass functions to be constructed by recursively inspecting the trees, a task of non-trivial computational cost even without leaf-list elision (Most Common Value elision was still enabled to reduce the memory consumption, see Section 6.1). This puts the GDyADt at a disadvantage, which achieved its highest rate of CI tests per second at LLT 0, namely 39.0 s${}^{-1}$, at the expense of consuming 2.7 GB of memory. As a secondary observation, the LLT parameter matters far less on ANDES than on the much smaller ALARM data set: the dynamic AD–trees consume similar amounts of memory regardless of the value of LLT, although a performance penalty of having a non-zero LLT value can be observed in the results.

Secondly, the GdcMI configuration benefits from the fact that our proposed optimization performs queries into simple data structures, as discussed in Section 5. Querying the JHT is significantly faster than recursing into the deep AD–trees. Moreover, the efficiency gain caused by the reusal rate of the joint entropy terms stored in the JHT was much more pronounced due to the far higher number of CI tests that had to be performed by IPC–MB on the ANDES data set. This reusal has kept the test rates of the GdcMI configuration very close to the test rates achieved on the ALARM data set, in spite of the ANDES data set containing six times more features. In addition, the memory consumption of the GdcMI configuration was around 3.6 times lower than the consumption of the GDyADt configurations. An eviction mechanism would have brought the memory consumption of GdcMI even lower, although this was not necessary in the experiment presented here.

## 8. Conclusions

The results of the comparative experiment presented in Section 7 show that our proposed optimization dcMI provides a significant efficiency gain when a large numbers of G-tests must be computed on the features of a data set. Such demanding situations are exemplified by the already efficient Markov blanket discovery algorithm IPC–MB, which relies on the G-test to determine conditional independence between the features of a data set. The dcMI optimization is also trivial to implement, requires no configuration parameters, and is widely applicable, although the comparative experiment presented in this article used IPC–MB specifically to evaluate the efficiency gains of dcMI.

Moreover, dcMI far exceeds the efficiency gains provided by either static or dynamic AD–trees, in spite of using far less memory. The most significant difference was observed on the ANDES data set, which contains 223 features, where dcMI outperformed the dynamic AD–tree by a factor of 20 to 21, depending on the number of samples in the data set instance, all the while consuming 3.6 times less memory. If the computationally expensive MCV elision had been disabled for the dynamic AD–tree, while also adding a configurable eviction mechanism to the data structures used by dcMI, the memory consumption ratio between the two optimizations would have been even higher. The results also show that, on the selected data sets, the dcMI optimization does not slow down significantly as the number of features in the data set increases. Even if the difference in feature count is a factor of 6, as it is between the ALARM and ANDES data sets, IPC–MB performed almost as fast on either data set when using dcMI.

However, it must be noted that dcMI is an optimization specific to computing mutual information, either conditional or unconditional. In its proposed form, it is not applicable in other cases. However, this specificity is not a restriction because mutual information is a fundamental concept in both information theory and in statistics. Indeed, dcMI greatly accelerates the computation of the *G* statistic in particular, but other applications that employ conditional mutual information may also benefit from it.

AD–trees are a more general optimization than dcMI because they have no built-in assumptions about their consumer. This fact makes them applicable in a wider variety of contexts, in spite of drawbacks such as memory consumption, complex querying logic, and becoming slower as the number of features in the data set increases. It must be emphasized that AD–trees allow for the construction of the contingency table of any feature in the data set, a universally useful application.

Improvements for computing the number of degrees of freedom of a G-test are also being considered. To accurately calculate degrees of freedom, linear algebra methods will likely have to be integrated into the dcMI optimization, with consideration for their computational cost and to the manner in which dcMI accesses the data set.

In the comparative experiment presented here, the significance of the G-test was uniformly set to 0.05, for consistency with the previous literature on which it is based. However, the effects of this uniformity will be examined further and the significance setting may be changed to an adaptive one, computed with respect to the test being performed or the data set itself.

## Figures and Tables

**Figure 1 entropy-23-01501-f001:**
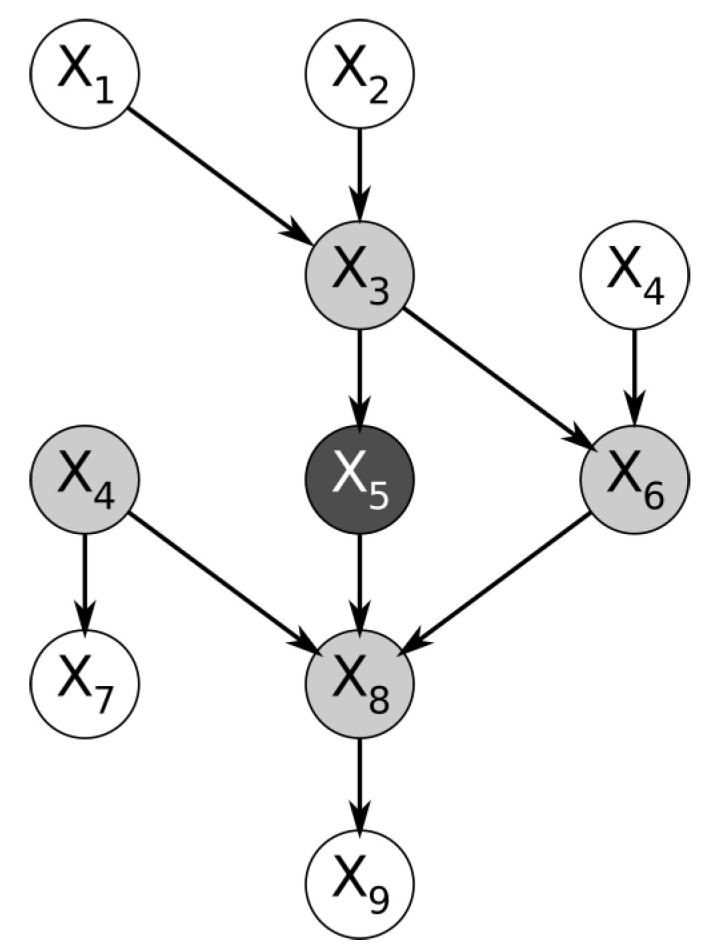
Example of Bayesian network; the Markov blanket of feature $X_{5}$ is highlighted.

**Table 1 entropy-23-01501-t001:** Pre-experimental build durations for static AD–trees, in seconds.

LLT	4000 Samples	8000 Samples	16,000 Samples
0	13.7	31.1	67.3
5	0.9	1.7	3.0
10	0.5	1.1	2.1

**Table 2 entropy-23-01501-t002:** Results of the ALARM subexperiment for 4000, 8000 and 16,000 samples. The table header also shows the amount of CI tests performed in each case. L00, L05, and L10 are abbreviations of LLT 0, LLT 5, and LLT 10, respectively.

	4000 Samples				8000 Samples				16,000 Samples		
	26,430 CI Tests				33,183 CI Tests				34,291 CI Tests		
**Configuration**	**Time**	**Rate**	**Mem**		**Time**	**Rate**	**Mem**		**Time**	**Rate**	**Mem**
	**(s)**	**(s** ${}^{-1}$ **)**	**(MB)**		**(s)**	**(s** ${}^{-1}$ **)**	**(MB)**		**(s)**	**(s** ${}^{-1}$ **)**	**(MB)**
GDefault	161	163.7	—		389	85.2	—		786	43.7	—
GStADt L00	24	1074.9	516.1		34	948.4	1123.4		38	894.0	2452.1
GStADt L05	57	459.2	34.1		95	348.5	54.8		123	278.4	87.4
GStADt L10	63	415.5	19.5		105	314.6	34.5		140	243.6	58.9
GDyADt L00	25	1044.4	14.6		37	892.5	28.6		40	838.5	49.6
GDyADt L05	58	453.7	12.6		96	345.4	25.3		127	268.5	45.2
GDyADt L10	66	400.0	11.1		106	310.9	23.0		143	239.0	41.7
GdcMI	11	2522.7	4.2		18	1815.4	4.6		32	1063.2	4.8

**Table 3 entropy-23-01501-t003:** Results of the ANDES subexperiment for 4000, 8000, and 16,000 samples. The table header also shows the amount of CI tests performed in each case. L00, L05, and L10 are abbreviations of LLT 0, LLT 5, and LLT 10, respectively.

	4000 Samples				8000 Samples				16,000 Samples		
	1,155,034 CI Tests				2,841,520 CI Tests				4,830,900 CI Tests		
**Configuration**	**Time**	**Rate**	**Mem**		**Time**	**Rate**	**Mem**		**Time**	**Rate**	**Mem**
	**(s)**	**(s** ${}^{-1}$ **)**	**(MB)**		**(s)**	**(s** ${}^{-1}$ **)**	**(MB)**		**(s)**	**(s** ${}^{-1}$ **)**	**(MB)**
GDyADt L00	6001	192.4	567.8		30,462	93.2	1287.2		123,680	39.0	2718.1
GDyADt L05	8783	131.5	563.3		50,171	56.6	1269.0		261,236	18.5	2677.9
GDyADt L10	10,398	111.0	559.0		64,255	44.2	1257.8		346,235	13.9	2653.5
GdcMI	481	2401.1	166.1		1665	1706.4	369.2		5752	839.7	731.0

## Data Availability

Data sets used in this study were automatically generated from publicly available data structures [30]. The experimental framework contains functionality to regenerate all the data sets exactly as used in the presented experiment [26], given the aforementioned data structures.

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
