# Peer review of "Accelerating Causal Inference and Feature Selection Methods through G-Test Computation Reuse"

_entropy, 2021, doi:10.3390/e23111501_

Round 1

Reviewer 1 Report

plese see the review attached

Reviewer 2 Report

Review of: 
Accelerating Causal Inference and Feature Selection Methods
through G-test Computation Reuse By Camil Bancioiu and Remus Brad 

This paper is very well written, interesting, and helpful for researchers in the area.  Markov blanket discovery and identification algorithm is  an important component is the construction of Bayesian Networks and other applications in statistical modeling.  In the last decades, several algorithms and methods have been published for this purpose. 

This paper describes, in detail, efficient implementations of the G-test (conditional independence test via likelihood ratio) using computationally efficient decomposition of the quantities of interest in reusable cross-entropy terms. 

As in many complex computational algorithms, attention to detail is of great importance, and this article does it with great care, all the way from high level descriptions to publicly available source code.     
In this respect, sub-section 5.1 (An Example of Reusing Joint Entropy Terms) is specially useful. 

Nevertheless, the construction of AD-trees does not receive in this article a description with the same level of care. I suggest the inclusion is Section 6 of a new sub-section giving toy examples of the growth of AD-trees used in the algorithm and experiments. Sure, one can always read the source code, but the presentation of well chosen toy examples is the best way to make this information understandable and easy to assimilate.  The authors did the effort in Sec.5.1 for joint entropy tables, they should do the same in section 6 for AD-trees. 

The availability of well implemented and documented source code for the algorithms and experiments described in this paper makes this contribution specially relevant. It certainly deserves to be published. I hope the authors take the time and effort (it will not be hard) to include the material I suggest. 

Reviewer 3 Report

Report on the paper "Accelerating Causal Inference and Feature Selection Methods through G-test Computation Reuse" submitted to Entropy  

Summary of the paper: The authors present a novel and remarkably efficient method of computing the so-called statistical G-test, which takes advantage of a connection with fundamental elements of information theory: the G statistic is rewritten as a sum of joint entropy terms, and its computation is decomposed into easily reusable partial results with no change in the resulting value. Because this decomposition allows for the intense reuse of these partial results, this method considerably increases the efficiency of applications that execute a number of G-tests on permutations of the same features, such as feature selection and causal inference applications. This method's efficiency is illustrated in an experiment with IPC-MB, an efficient Markov blanket discovery process that can be used as a feature selection algorithm as well as a causal inference method. When the G-test is calculated with the proposed method, the results indicate outstanding efficiency benefits for IPC-MB, not only when compared to the unoptimized G-test, but also when compared to IPC-MB++, a variant of IPC-MB that is augmented with an AD-tree, both static and dynamic. Even though this proposed method for computing the G-test is presented in the context of IPC-MB, it is not restricted to IPC-MB in particular, nor to feature selection or causal inference applications in general, because it focuses on the information-theoretic concept that underpins the G-test, conditional mutual information. Because of this, the authors claim that it has a wide range of applications in data science.

  Evaluation: The paper contains an interesting review of the subject and some interesting material. However, the paper is very hard to follow and the contributions are not strong enough to justify such a massive document. In my opinion, the paper must be rewritten and drastically reduced, with a focus on the real contributions. The organization of the paper is very unclear, and the text is too expansive and disproportionated in view of the importance of the findings. I suggest a resubmission with a total rewriting of the paper.   

Reviewer 4 Report

Please see the comment in the attached file.

Reviewer 5 Report

Review of "Accelerating Causal Inference and Feature Selection Methods through G-test Computation Reuse" by Bancioiu and Remus (2021)

The authors presented an interesting paper about G-test for causality inference. Some datasets example are also well illustrated by the method. I think that paper could be published; however, I detected several lacks: 

1. Introduction. Authors could support the ideas by inserting in L42: "...about the process that produced the data (Contreras-Reyes and Hernández-Santoro, 2020)", in L133: "... of the data set, one at a time (Sun and Bollt, 2014)". Also, authors could also explain that exists other methods for independence test based on mutual information using bayesian nonparametric estimation (Al-Labadi et al., 2021).

2. For continuity, merge paragraphs in L24-25, 71-72, 109-110, 119-120, 125-126, 156-157, 307-308, 337-338, 759-760, 808-809, and 885-886.

3. Figure 1 is not referenced in the text.

4. L256: $H_0:\,\mathbb{P}()...$  . Replace "Pr" by \mathbb{P} (in latex) in all manuscript.

5. L271 and 283-284 : Please give the degrees of freedom for chi-square distribution. Specifically, give more details about DoF computation given in [14] (number of independent standard normal variables). It could depends on distribution (Pr) used? what's happen if x,y,z are bernoulli of Poisson distributed? Note that you highlight as further research this issue in lines 909-912.

6. L275: "human experimenter" <-> "researcher".

7. Given that manuscript is too large, I think that example of section 5.1 could be removed (unnecessary) or appears in Appendix.

8. Both equations of page 11 are already defined. Perhaps section 5.2.1 could be removed. The most important part appears in 5.2.2. Also in 5.2.3, Equation 8 is repeated, please summarize the results by calling the equation.

9. L433(+1), 434(+1), etc...: "For the case" <-> "Case".

10. L747, 759: "online repository" <-> "R package". L748: "larkparser Python package".

11. L763: "7.3" <-> "7.2.1", "7.4" <-> "7.2.2".

12. L903: delete "9. Further Work", and delete lines 918-921 (unnecessary).

References:

Al-Labadi, L., Asl, F. F., Saberi, Z. (2021). A Test for Independence Via Bayesian Nonparametric Estimation of Mutual Information. Can. J. Stat., in press. DOI: https://doi.org/10.1002/cjs.11645

Contreras-Reyes, J.E., Hernández-Santoro, C. (2020). Assessing Granger-causality in the southern Humboldt current ecosystem using cross-spectral methods. Entropy 22(10), 1071. 

Sun, J., Bollt, E. M. (2014). Causation entropy identifies indirect influences, dominance of neighbors and anticipatory couplings. Physica D 267, 49-57. 

Round 2

Reviewer 2 Report

From the authors response to my review: 

>Due to time constraints, the example presented in Appendix C does not cover the optimizations speci c to AD-trees,
namely Leaf-List elision and Most Common Value elision, which greatly reduce the size of the tree at the cost of more
complex querying. 

The strongest point of the article, in my perspective, is, together with the availability of source code, the clear and detailed presentation of all details of the algorithm and its implementation. 

Hence, I urge the authors to complete their work in this article, and complete a detailed description of AD-trees optimization. 

This is Not an article for those who want a quick overview of key ideas, and a general understanding of how they works. This is an article to be read by someone willing to understand ``exactly how it is done'', how to implement the algorithm in all its details. Hence, it would be a shame to provide such detailed explanation for 80% of the work, and leve the reader guessing on how to complete the journey (oder than reading the source code). 

If the overall length  of the article is an issue (I do Not think it is), make some detailed descriptions appendices to the main text. Nevertheless, I must stress, both to authors and to the editors: Articles explaining detailed implementations of complex algorithms are, by their very nature, Long. A faint hearted reader may get easily impatient or bored -- that is irrelevant. The authors should no compromise by taking shortcuts in hope of not upsetting those fain hearted readers -- they will not be interested in this article anyway.

Those readers that really need or want to understand in detail how it is done, and take to effort to read the article all the way, should not be disappointed by an incomplete description. Please finish the work. 

Finally, let me congratulate the authors by a work well done. Many times, research articles presenting flashy new ideas are highly praised, while the efficient implementation of complex algorithms get little attention. Nevertheless, exactly those efficient implementations are the ones used in the practice of science. Hence they deserve to receive appropriate attention in the scientific literature. 

I suggest the authors add a note to the source code (or its derived application softwares) asking any researcher that uses it to cite the final version of this article in all their research articles. This would be a fair recognition for the authors work. Moreover it would avoid a frequent problem in scientific research, namely, the use of black box, or poorly, or incompletely understood software tools. 

Reviewer 3 Report

This revised version is much better. I approve this version, and recommend it for publication. 

Author Response

Thank you kindly for recommending the article for publication.

Reviewer 4 Report

The authors have made the necessary changes to the manuscript.

Author Response

Thank you for accepting the revision.

Reviewer 5 Report

In this 2nd review, I can see than only one of my question has not been replied in well form, and is related to the degrees of freedom (DoF) for chi-square distribution. I understand that the steps appear in a reference, but the author could give a effort to explain with more detail this important issue. The DoF are necessary to performs an statistical test and give a decision. Authors could include the steps to obtain DoF in Appendix (as well as they included material in previous review). 

Round 3

Reviewer 5 Report

No further comments. All of my previous suggestions/comments have been addressed by the authors.